# Natural Products That Changed Society

**DOI:** 10.3390/biomedicines9050472

**Published:** 2021-04-26

**Authors:** Søren Brøgger Christensen

**Affiliations:** The Museum of Natural Medicine & The Pharmacognostic Collection, University of Copenhagen, DK-2100 Copenhagen, Denmark; soren.christensen@sund.ku.dk; Tel.: +45-3533-6253

**Keywords:** malaria, quinine, chloroquine, artemisinin, onchocerciasis, ivermectin, moxidectin, cancer, vincristine, vinblastine

## Abstract

Until the end of the 19th century all drugs were natural products or minerals. During the 19th century chemists succeeded in isolating pure natural products such as quinine, morphine, codeine and other compounds with beneficial effects. Pure compounds enabled accurate dosing to achieve serum levels within the pharmacological window and reproducible clinical effects. During the 20th and the 21st century synthetic compounds became the major source of drugs. In spite of the impressive results achieved within the art of synthetic chemistry, natural products or modified natural products still constitute almost half of drugs used for treatment of cancer and diseases like malaria, onchocerciasis and lymphatic filariasis caused by parasites. A turning point in the fight against the devastating burden of malaria was obtained in the 17th century by the discovery that bark from trees belonging to the genus *Cinchona* could be used for treatment with varying success. However isolation and use of the active principle, quinine, in 1820, afforded a breakthrough in the treatment. In the 20th century the synthetic drug chloroquine severely reduced the burden of malaria. However, resistance made this drug obsolete. Subsequently artemisinin isolated from traditional Chinese medicine turned out to be an efficient antimalarial drug overcoming the problem of chloroquine resistance for a while. The use of synthetic analogues such as chloroquine or semisynthetic drugs such as artemether or artesunate further improved the possibilities for healing malaria. Onchocerciasis (river blindness) made life in large parts of Africa and South America miserable. The discovery of the healing effects of the macrocyclic lactone ivermectin enabled control and partly elimination of the disease by annual mass distribution of the drug. Also in the case of ivermectin improved semisynthetic derivatives have found their way into the clinic. Ivermectin also is an efficient drug for treatment of lymphatic filariasis. The serendipitous discovery of the ability of the spindle toxins to control the growth of fast proliferating cancer cells armed physicians with a new efficient tool for treatment of some cancer diseases. These possibilities have been elaborated through preparation of semisynthetic analogues. Today vincristine and vinblastine and semisynthetic analogues are powerful weapons against cancer diseases.

## 1. Introduction

Until the end of the 19th century only natural products and minerals were available as drugs. Nature today still continuously surprises by offering molecules outside the scope of human creativity to give drugs enabling healing of previous incurable diseases [1,2,3]. Natural products still are the basis for half of all new drugs, either as the parent natural products or optimized derivatives [3,4]. Marine organism and cultures of fungi including endophytes have added new possibilities for finding miraculous natural products [3,5,6]. Until the 20th century malaria was an almost worldwide burden causing death and disability. Quinine and artemisimin made treatment of malaria successful [7,8,9]. Onchocerciasis (river blindness) prevented cultivation of riverbanks particularly in Sub-Sahara Africa but also in Latin-America. Lymphatic filariasis made life miserable for millions of people in Africa and Latin America. The natural product ivermectin enabled treatment of onchocerciasis, lymphatic filariasis and other parasitic diseases [10]. Fast proliferating cancer diseases had a severe death rate until the middle of the 20th century, when the vinca alkaloids were discovered. Vincristine, vinblastine and semisynthetic analogues, became efficient drugs for treatment of some of these cancer diseases [8,11]. These drugs are examples of drugs that changed the society and they will be mentioned in the present review. Many more natural products that also had an impact on the society e.g., antibiotics such as penicillin and chemotherapeutics like taxols have been omitted because of limited space and time.

Some natural products are only found in trace amounts in few organisms. Today we are able in some cases to solve the problem of sustainable supply by total synthesis of the compounds. Semisynthesis enables design and preparation of more efficient drugs by modifying the structure of the parent natural product or by making prodrugs [4]. This only has been possible for the last 150 years. Actually vitalism denied the possibility of synthesis of organic compounds since a theorem in this philosophy claims that only living organism can form C-C bonds [12]. The first evidence of the limitations of this paradigm came when Wöhler in 1828 published the synthesis of urea by dissolving silver cyanate or lead cyanate in aqueous ammonia [13]. Since this reaction did not involve formation of C-C bonds this might appear a poor argument. A convincing counterevidence was obtained when tetrachloromethane (**2**) was converted in tetrachloroethene (**3**), which then was converted into acetic acid (**5**) via trichloroacetic acid (**4**). Since tetrachloroemethane was obtained from carbon disulfide (**1**) prepared from charcoal and sulfur this proved that C-C bonds could be formed in vitro using only inorganic reagents (Scheme 1) [14]: 

The art of organic synthesis was developed during the 19th, 20th and 21st century and today very complex organic molecules have been prepared by total synthesis. Total synthesis is a procedure for preparing highly complex natural products starting with simple commercially available materials [15,16,17,18]. Acetylsalicylic acid (**6**), an analogue of salicin the antipyretic principle isolated from willow barks, was synthesized in 1859 but not marketed under the name of aspirin before 1899 [19,20]. Phenazone (**7**) was synthesized in 1887 and marketed as antipyrin in 1889–1890 during an influenza epidemic (Figure 1) [21]. 

Even though synthetic chemistry has made impressive progress, many natural products either are so complex or so easily available that the compound isolated from biological material or isolated from cell cultures is still used in pharmaceutical formulations. This is the case for e.g., taxol, morphine, codeine, vincristine, vinblastine and quinine [22]. One of the most complicated natural products prepared by synthesis is eribulin [23]. Even though a procedure for total synthesis of a natural product has been developed this frequent is so complicated and with such a poor yield that it is not economically feasible for production of the compound on kg scale. A procedure for synthesis of quinine was described in 1944. This achievement though highly praised during the war never made synthetic quinine available, because the many steps in the synthesis. It still is debated whether the synthesis from 1944 leads to quinine. An undisputed total synthesis of quinine was not developed until 2001, and even this synthesis is not feasible for production of quinine in kg scale [24].

The most important of all natural products, defined as compounds produced by a living organism, is dioxygen, normally just mentioned as oxygen. Dioxygen is formed by photosynthetic cleavage of water (Scheme 2) [25]. Dioxygen was only present in trace amount in the atmosphere of the earth before living organism developed photosynthesis and, thus, must be considered a natural product [25,26].

## 2. Malaria

Malaria is a disease caused by five different protozoan parasites of the genus *Plamodium* (*P. malaria* causing quartan malaria, *P. ovale* causing ovale tertian malaria, *P. vivax* causing benign tertian malaria, *P. knowlesi*, which only recently has been shown to infect humans, and *P. falciparum* responsible for the majority of fatalities caused by malaria) [7]. The life cycle of the parasite is depicted in Figure 2.

Protozoan parasites are parasites living inside the cells of the host. From the *Plasmodium* genus the most deadly parasite is *P. falciparum* causing malignant tertian malaria, also known as cerebral malaria. This species is only transmitted in the tropics. Two studies formed the basis for our understanding of the disease, the first was the discovery of parasites in the erythrocytes of malaria patients by Alphonse Laveran in 1880 and the second the role of *Anopheles* mosquitoes as vectors described by Ronald Ross in 1897 [7,27,28]. Both researchers were awarded the Nobel Prize in 1907 and 1902, respectively. In contrast to *P. falciparum* the other parasites are also infectious in subtropical and temperate areas. About year 1900 malaria had its maximal distribution reaching latitudinal extremes of 64° norths and 32° south encompassing all continents Europe, the Americas, Asia and Australia. In year 2004 the area in which humans were at risk of being infected with malaria was decreased from 53% to 27% of the Earth’s land surface [29]. In the period 2010 to 2018 the estimated cases of malaria has decreased from 251 million to 228 million and the number of deaths from 585,000 to 405,000, which still are scaring high numbers [30]. Previous the presence of *P. falciparum* meant that Europeans were prevented from entering tropic Africa, because of the mortality from cerebral malaria. An about seven times higher death rate for French and British soldiers were observed in the tropic African colonies in the first half of the 19th century than seen elsewhere [31]. In 1805 a party of 44 Europeans sailed up the river Niger but only five survived the journey [32] illustrating the mortality of malaria. No doubt the high mortality rate prevented colonization of Africa (Figure 3). 

Even though mortality of malaria in Europe was not comparable with that in Africa, the disease still was a severe burden. Malaria (known as ague or intermittent fever, in Danish: *koldfeber*, cold fever) was prevalent in Southern England in Kent, Essex and parts of London in the 17th century. Rome was also reputed for malaria [32].

Malaria and a number of other diseases well known in Europe, Asia and Africa such as smallpox, influenza, measles and yellow fever were no burdens for the pre-Columbian American population [33]. Consequently the endemic population had no immunologic defense. The epidemic infections brought to America by infected Spaniards led to demographic falls. The Indian population on the island Espanola decreased from 3,770,000 in 1496 to 125 in 1570. Similar catastrophic falls were observed in Mexico and Peru [33]. Mating between Spaniards and Indians created Mestizos, who were less sensitive to the diseases and soon outnumbered the original population [33]. 

### 2.1. Drugs to Treat and Prevent Malaria

#### 2.1.1. Cinchona Bark

After the conquest of the Inca Empire in South America the Jesuit monks followed the conquistadores as missionaries. Jesuits, who were the apothecaries at that time, noticed that the Incas used the bark of cinchona trees when working in cold streams to prevent shivering [34]. They hypothesized that the bark also might prevent shivering provoked by intermittent fever (malaria). A source mentions that the first European patient treated with cinchona bark was a Spaniard treated in the territory of Loja, in southern Ecuador in 1631 [33,35]. A successful outcome of this case study encouraged the Jesuits monks to export the bark to Europe [32,36]. In 1632 Jesuits send the first batch to Europe [37]. The bark was imported to Europe under different names, Jesuit bark acknowledging the Jesuits, who discovered the healing effects, Cardinal’s bark, honoring Cardinal Lugo, or Peruvian bark. The genus name *Cinchona* was invented by Linnaeus in his Systema Naturae (1742) [37]. The genus was grouped in the plant family Rubiaceae (Figure 4)

Already in 1649 a recipe for treatment of malaria, Schedula Romana, summarized trials carried out using cinchona bark [34,35,37]. Protestant Europe only reluctantly used the Catholic drug. Another obstacle was the doctrine of Hippocrates that well-being depends on the balance between the four humours or liquids: yellow bile, black bile, phlegm and blood [12]. The bark did not provoke bleeding, vomiting or in other ways affect the balance between the four humours and consequently, it could not have a healing effect [37]. Robert Talbor, a former apothecary and therefore not so concerned about this doctrine, became famous by successfully curing King Charles II of England and the Dauphin and Dauphine of France for malaria. In 1677 cinchona bark was included in the London pharmacopoeia as Cortex Peruvianus [37] (Figure 5). 

From a modern scientific standard it is amazing to realize that the bark was included in a pharmacopoeia even though the botanical origin was not known until 1738 [34,37]. The missing knowledge of the mother plant facilitated trading with ineffective and even poisonous barks from other trees [34].

Cinchona bark was used so extensively by the Europeans that the source was in danger of becoming extinct by the middle of the 19th century [36]. In spite of an embargo a Dutchman claiming to be a tourist in 1852 smuggled seeds and seedlings from the South Americans cinchona territories to Java, Indonesia (Dutch East Indies), and established a plantation [39]. The plantations on Java became very successful so in 1924 Java had almost monopoly in production of *Cinchona* bark [40]. British botanists organized expeditions in South America to get the plant material to India [41]. These are early examples of what today is now known as bio-piracy [42,43]. In favor of these illegal exportations it might be argued that otherwise the *Cinchona* sources probably would have been overexploited leading to extinction in South America. 

In the Danish Pharmacopoeia from 1805 are listed four different cinchona barks: Cortex Carabaeus harvested from *Cinchona caribaea*, which is a synonym of *Exostema caribaeum* (Jacq) Schult (Rubiaceae). The bark is known to be extremely bitter because of the presence of 4-phenylcoumarins, but cinchona alkaloids have not been found [44]. Cortex Peruvianus, harvested from *Cinchonae officinalis* L., Cortex Peruvianus Flavus (yellow or royal bark) and Cortex Peruvianus Ruber (red bark). No species is mentioned for the latter two but later they have been assigned to *C. calisaya* Wedd (yellow bark) and *C. pubescens* Vahl (red bark) [45]. An investigation of historical samples of cinchona barks have revealed a variation in quinine content from only trace amounts to 5% and no species stands out by possessing a high amount of quinine [46]. Similar analyses of species of *C. calisaya* collected in the field at different locations in Bolivia reveals variations in quinine content in the bark from trace amounts to 25% [45]. These findings confirmed the results of analysis from 1945 revealing variation in the quinine content in *C. officinalis* and *C. pubescens* species collected in Ecuador, Bolivia and Peru from trace amounts to 7% [47]. Such inhomogeneity must have made correct dosing of the bark a problem considering the poor analytical techniques present in the 18th and 19th centuries and considering the limited therapeutic window for quinine [7]. In the Danish Pharmacopoeia from 1907 only Cortex Chinae is mentioned, which is defined as bark originating from different preferential cultivated *Cinchona* species. The bark must contain at least 4% of alkaloids and at least 1% of quinine. Quinine chloride was included in the Danish Pharmacopoeia from 1893.

#### 2.1.2. Quinine

In 1820 Pelletier and Caventou succeeded in isolating an active principle from *Cinchona* bark [34,48]. The compound was named quinine (Figure 6, **8**). To isolate quinine a suspension of ground bark slurried in water mixed with slaked lime was extracted with an organic solvent like hot toluene. The organic layer was extracted with diluted sulfuric acid and upon concentration of the aqueous phase quinine sulfate crystallized [37,39]. This procedure enabled large scale production. Already early comparisons of the use of *Cinchona* bark or quinine sulfate revealed advantages of using the isolated active principle [34]. Quinine sulfate was given in a pill in a dose of 1 g each day. In contrast several grams of the crude bark had to be swallowed and even though wine was offered to make swallowing easier it must have been unpleasant. Decoctions and tinctures of the bark were also used, but probably not standardized. 

In addition to being easier to swallow, the most important consequence of the use of the isolated compound was that it enabled administration of well-defined doses. Considering the variation of quinine content in the different barks and taking into account the analytical techniques available in the early 19th century the use of purified quinine facilitated correct dosing, even though a skilled apothecary probably would have a good feeling of the quality of the bark [34,35]. In Denmark a price for quinine sulfate was fixed in 1840 by the government (Interims Taxt 1840).

Inspection of the relocation costs, defined as the increased mortality among soldiers by relocation to tropical areas (“white man’s grave”), reveals a dramatic fall around 1840. This increased survival has been related to the use of purified quinine [31,49] instead of cinchona bark. During the 19th century the mortality of sailors on the naval operations stopping slave trade was severely reduced after introduction of quinine for prevention and cure [31,49]. Earlier attempts using cinchona bark never gave convincing results [49]. It is interesting to notice that the Berlin Conference regulating the European colonization and trade in Africa took place 1884–1885 after the prophylactic use of quinine had been generally accepted. Quinine also played an important role during the American Civil War. The war might have ended with a victory to the Union troops already in the first year had malaria and typhoid fever not made the soldiers unable to fight [36].

In 1928 the total consumption of quinine in India was estimated to be 160,000 lbs (72 tons) [50]. In 1924 the world consumption of quinine sulfate as 500 ton, of which 95% was produced from 10,000 ton of bark grown on Java giving the colony an income of 16–22 million £. India was the second most important supplier for 10 million US$ corresponding to 25 ton of quinine sulfate [36,39]. From the mid-19th century until 1940 quinine was the standard treatment of malaria [36]. Even though quinine had been used extensively since Pelletier and Cavantou isolated the compound in 1820 the first suggestion of the constitution was published by Paul Rabe in 1908 [51] and the relative configuration in 1950 (Figure 6, **8**) [52]. In spite of the missing knowledge about the structure, salts of quinine had been used as drugs since 1820.

#### 2.1.3. Quinine Resistance

Resistance to quinine appears sporadically, but failure of treatment is only seen in Southeast Asia and New Guinea [53].

#### 2.1.4. Synthetic Malaria Drugs

An early attempt to produce synthetic quinine was made by William Perkin in 1856 based on a hypothesis which today appears naïve. The attempt did not lead to quinine but instead the dye mauve. After Queen Victoria appeared public dressed in mauve dyed robes the dye became an enormous commercial success [54]. 

Paul Ehrlich in 1891 treated a malaria patient with methylene blue (Figure 7, **12**). However, this dye never became useful in the clinic as an antimalarial drug. It was, however, used as a template for developing chloroquine (**13**) [55,56]. The first synthetic antimalarial used in the clinic was pamaquine (plasmoquine, **16**) from 1924, which was followed by mepacrine (atabrine, quinacrine, **17**) in 1932 and resochin (chloroquine) in 1934 (**13**) [7]. Atabrine was first marketed in Germany in 1932 [57]. Chloroquine was evaluated to be too toxic for clinical use and consequently efforts to make a less toxic analogue were undertaken to give the 2-methylated analogue sontochin (**14**) (1936) [57]. 

#### 2.1.5. Malaria during the Pacific War 

It is generally known that after the Japanese bombing of Pearl Harbor in 1941 the USA entered World War 2. Less known is that the mortality due to malaria on the Pacific Islands was far beyond that caused by Japanese troops [36]. The Japanese occupation of Dutch East Indies, in particular Java, meant an interruption of the supply of sufficient amounts of quinine to the Allies. Consequently intensive research programs were initiated in order to find alternative antimalarial drugs [57]. As a first attempt totaquine, a mixture of alkaloids extracted from *Cinchona* bark was investigated [57]. However, the amounts of *Cinchona* bark available could not meet the demand. Like the Allies the Japanese used the synthetic primaquine (**15**) for prophylaxis but the Japanese in insufficient doses [36]. In spite of these precautions malaria caused unacceptable losses. A group at the National institute of Health was organized in 1939 to develop alternative antimalarial drugs. The group organized a Committee on Medical Research, which collaborated with universities, industry and private individuals, the US Army and Navy and appropriate scientists in Britain and Australia. The collaboration resulted in synthesis of about 16,000 compounds. Some 80 compounds were selected. The most promising compound turned out to be chloroquine (**13**) already patented, but discarded by Bayer in Germany [57]. The 4-methyl derivative sontochin (**14**) was for a period preferred to chloroquine. 

Plantations were started in Tanzania, Kenya, Cameroon and Rwanda. Also a joint venture between USA and Guatemala resulted in 400 ha with 1.75 million trees by mid-1948 [39]. These new plantations, however, did not solve the problem with quinine supply during the war since the amount of bark which can be harvested from a tree is very dependent on the age of the tree. From a four year old *C. ledgeriana* 0.25 kg, from an eight year old tree 4 kg, from a 15 year old 10 kg and from a 25 year old 20 kg of dried bark can be harvested [40]. After harvest of the bark the trees are covered up enabling continued growth.

Woodward in 1944 published a total synthesis of quinine but even though it created hope for overcoming the Japanese disruption of quinine supply during the Second World War, the many steps in this procedure, however, prevented its commercial use and even today quinine is still produced from Cinchona bark [54].

#### 2.1.6. Malaria Control 

Intensive studies performed by the Allies established in 1944 that chloroquine was a safe antimalarial. By combination of chloroquine and insecticides like DDT programs to eliminate malaria were initiated [58,59,60]. In Africa chloroquine became a pillar for malaria eradication [61,62]. In the late 1950s chloroquine-resistant parasites developed in six independent regions in India, Thailand, Indonesia, New Guinea, Venezuela and Guyana and spread over most of the areas infected with malaria [63]. As a consequence malaria re-emerged India [58], Kenya [64], and Africa [61,62]. 

A contribution to the decreased chloroquine sensitivity is mutations in the parasite genome to form a *pfcrt* gene. This gene encodes the *Plasmodium falciparum* chloroquine-resistant transporter (*Pf*CRT) a 424 amino acid protein with 10 transmembrane helices that facilitates transport of chloroquine away from ferriprotoporphyrin IX [59,63,65,66]. Today resistance has made chloroquine obsolete to control *P. falciparum* infections; but it still is used against *P. vivax*. Even though the use of chloroquine today is limited, the previous importance of chloroquine to reduce the burden of malaria should not be underestimated [60]. In the interim from appearance of chloroquine resistant-parasites to the appearance of the artemisnins synthetic a number of drugs were used to control malaria. Among these drugs are mefloquine (Figure 8, **17**), halofantrine (**22**), lumefantrine (**23**), primaquine (**15**), atovaquone (**20**), sulfadoxine (**21**), tetracycline (**25**) and proguanil (**18**), a prodrug which in the liver is transformed into the active drug cycloguanil (**19**). *Plasmodium* parasites host an endoparasite, the apicoplast. The apicoplast has a bacterial ancestry. Consequently the naturally occurring antibiotic tetracycline, which binds to the ribosome of bacteria, can be used for treatment of malaria. Terpene and type II fatty acid biosynthesis also proceed in the apicoplasts. Blockage of this system will not kill the mother parasites but daughter parasites with malfunctioning apicoplasts are affected [67]. Tafenoquine (**24**) was approved for treatment of malaria in 2018 [68,69]. Attempts to use the natural product fosfidomycin, which targets the deoxyxylose pathway in the apicoplast, have not lead to clinical useful drugs [70,71,72].

Ferroquine (Figure 9, **26**) possesses the same pharmacophore as chloroquine but contains a ferrocene moiety, which prevents the compound from being a substrate for the *Pf*CRT transporter [73,74]. The compound was brought to clinical trials II [75] but has not been approved as a drug by U.S. Food and Drug Administration (FDA) USA.

#### 2.1.7. Mechanism of Action of Quinolines

Both quinine and chloroquine are amines and consequently they will be protonated in acidic media. The target molecule of both compounds is heme (ferriprotoporphyrin IX, Figure 10, **29**). 

Heme is formed in the food vacuole of the parasite when hemoglobin is digested to supply the parasite with amino acids. The pH of the food vacuole varies from 5.0 to 5.2, 1 to 2 units lower than that of the cytosol in the parasite. Consequently the neutral base will diffuse into the food vacuole where it accumulates after protonation [76]. After cleavage of hemoglobin the ferroprotoporphyrin **28** remains. In the ferrprotophorphytin the Fe^2+^ is quickly oxidized to Fe^3+^ affording ferriprotoporphyrin IX (**29**), which will provoke formation of reactive oxygen species unless inactivated [60]. Under physiologic conditions the ferriprotophorphyrin IX will polymerize to hemozoin (**30**), the characteristic malaria pigment seen in infected erythrocytes as dark spots. In the presence of chloroquine or quinine this polymerization is prevented, leading to the death of the parasite [76,77].

Even though many isoquinoline antimalarial drugs are chiral, resolution is not performed. The biological target molecule for these compounds ferriprotoporphyrin IX, (**29**) is achiral meaning that the two enantiomers have the same affinity. In contrast the volume of distribution, clearance and adverse effects do depend on the stereochemistry [78].

#### 2.1.8. Artemisinins

According to a legend a meeting between Mao Zedong and Ho Chi Minh led to a secret research program to find a new drug for controlling malaria among the soldiers fighting for North Vietnam in the Vietnam civil war [79]. The program was initiated in 1967 as Project 523. In contrast to the program set up by the Allies in 1939 this project resulted in a novel class of antimalarial drugs. The program was managed by the Chinese chemist You-You Tu, who in 2015 was awarded the Nobel Prize for the outcome. A starting point was taken in traditional Chinese medicine. Screening 2000 herb preparations afforded 640 hits. An extract from *Artemisia annua* L. (Asteraceae) showed promising but non-reproducible antiplasmodial effect (Figure 11).

Studies of the ancient Ge Hong’s A Handbook of Prescriptions for Emergencies (340 AD) revealed that extraction was performed by wringing out the juice of the plant in cold water. Inspired by this information the extracts were made at low temperature and an active extract was obtained. In 1972 a colorless crystalline substance was obtained and named quinghaosu, meaning the active principle in quinghao, the Chinese name for *A. annua*. Today the name artemisinin (Scheme 3, **31**) is preferred, at least outside China. Reduction of the lactone to an acetal afforded a mixture of the two dihydroartemisnins **32a** and **32b**, which turned out to be a better drug than the mother compound [80,81]. Artemether (**33**) and arteether (**34**) are lipid soluble. Artesunate (**35**) is soluble in water. 

The first report in English of this amazing study appeared in 1979, some years after the end of the Vietnam War [82]. Later it has been realized that the structure of artemisinin had been previously published by Stefanivic et al. in 1972 at a Symposium on the Chemistry of Natural Products in New Delhi. The compound was named arteannuin. Unfortunately this group never realized the importance of the discovery [83]. This story emphasize that you do not become an outstanding scientist just because you are lucky. You have to realize that you have been lucky.

#### 2.1.9. Mechanism of Action of Artemisinins

The pharmacophore of artemisinin is the 1,2,4-trioxane ring containing an endocyclic peroxide. Removal of this peroxide means loss of all antiplasmodial activity [77]. Whereas the artemisinins kills plasmodium parasites in low nanomolar concentration they are only toxic towards mammalian cells including fast proliferating cancer cell lines in micromolar concentration. Erythrocytes infected with plasmodium parasites have a rapid selective uptake of artemisinins in contrast to non-infected erythrocytes. After uptake the drug can be detected in the cytosol and the food vacuole. The Fe^2+^ in ferroprotoporphyrin (Scheme 4, **27**) might interact with the peroxide bridge in the artemisinins to give a reactive C-radical, which may alkylate the porphyrin skeleton preventing it from precipitation as heme (**28**) and consequently enabling the formation of ROS fatal for the parasite [77,84]. 

In addition to ferroprotoporphyrin (**27**) alkylation, the formed C-radicals also alkylate proteins and lipids, causing extensive cellular damage. The buildup of damaged proteins induces ER stress and attenuation of translation, which affords a lethal level of ubiquitinated proteins. Eventually merozoite death occurs. This mechanism of action differentiates artemisinins from other malaria drugs, which typically have a well-defined target [85]. The selectivity of the artemisinins originates in the Fe^2+^ needed for generation of the C-radicals.

#### 2.1.10. Clinical Use of Artemisinins

Poor absorbance of artemisinin (**31**) makes artesunate (**35**) the preferred drug among the arteminsinins. Artemether (**33**) and artesunate (**35**) are both prodrugs of dihydroartemisnin (**32a** and **32b**). Artesuante is soluble in water and consequently preferred. However, artemether can have advantages by rectal administration. Because of very rapid killing of parasites the artemisinins are very efficient for treatment of falciparum malaria [7,81].

In the presence of dihydroartemisin in the blood the number of parasites will be reduced 10,000 fold for each asexual cell cycle (approximately 36 h), whereas antimalarial antibiotics such as tetracycline only reveal a 10-fold reduction of parasitaemia. This rapid reduction is in particular advantageous for treatment of falciparum malaria since death might occur within hours after the first symptoms appear [7]. A drawback of the artemisinins is a very short half-live of about 1 hour. Rapid elimination from the blood also increases the risk of recrudescence [7,85]. In contrast a compound like chloroquine has a half-life of 30–60 days, proguanil 16 hours, primaquine 6 hours and lumefantrine 86 hours [7]. The short half-live of the arteminisins creates a risk for a long period with low blood level and consequently a high risk for development of resistance. To avoid this risk artemisinin combination therapy (ACT) is recommended meaning that artemisinins are given with drugs possessing a long half-live such as sulfadoxine (**21**)-pyrimethamine, mefloquine (Figure 12, **38**), amodiaquine (**39**) or lumefantrine (**23**) [7]. ACT has now become first choice for treatment of uncomplicated falciparum malaria [85]. The use of artemisinins has thus become an important tool for malaria control. 

#### 2.1.11. Resistance toward Artemsinins

In spite of the precautions artemisinin resistance defined a decreased sensitivity for artemisinins have occurred in South East Asia from China to India, in Equitorial Guinea and Uganda in Africa [85]. Considering the burden of malaria in Africa the development on this continent is particularly alarming. Parasites resistant to artemisinins are characterized by the presence of *PfK13* mutations. The Kelch 13 proteins are substrate for an E3-ligase, which bind to phosphatidylinositol-3-kinase and marks it for ubiquitination and eventually cleavage. Mutations in Kelch13 decrease the ubiquitination and consequent an increased level of Kelch13, thereby an increase level of phosphatidylinositol-3-kinase and finally of the product phosphatidylinositol-3-phosphate. An increased level of this lipid makes the parasite less sensitive to oxidative damages [85]. In spite of the decreased response ACT still is the first choice for treatment of uncomplicated falciparum malaria. 

#### 2.1.12. Sustainable Supply of Artemisnin

WHO has recommended ACT as the drug of first choice for treatment of malaria. The recognition has created an increased demand for the agents. The current source is cultivated *A. annua*. Attempts to increase the artemisinin yield have afforded two cultivars with about 2% contents. Unfortunately some of the established varieties are not stable over generations. Attempts to produce artemisinin by bioengineering using cell cultures have not been successful at the present. Heterologous production of precursors which by semi-synthesis can be converted into the target compounds has only been partially commercially successful [86].

#### 2.1.13. New Drugs for Treatment and Prophylaxis of Malaria

The Medicine for Malaria Venture, a non-profit, private-public partnership was established in 1999. The current portfolio contains a number of promising drug candidates like arterolane (OZ277, Figure 13, **40**), artefenomel (OZ439, **41**) and ferroquine (**30**) [73,74]. Artefenomel (**40**) and arterolane (**39**), like the artemisinins, contain a 1,2,4-trioxane ring and has the same mechanism [87]. The advantages of these drugs are longer half-lives in the body, 4 hours for arterolane [87] and 23 hours for artefenomel [73,87] in contrast to the 1 hour half-life of artemisinins. Long time exposure to the drugs is expected to make Kelch13 mutated parasites more sensitive [73]. Arterolane (**40**) is presently used clinically in combination with primaquine in India [73,87]. Other 1,2,4-trioxanes are also tested. 

Both ferroquine (**26**) and artefenomel (**41**) are synthetic drugs, which can be prepared on a large scale. 

Other drugs under development and the status of the development by Medicine for Malaria Venture can be seen on their homepage https://www.mmv.org/research-development/mmv-supported-projects, accessed on 1 April 2021. The ultimate goal of the project is to find at drug that can cure malaria in a single dose.

## 3. Onchocerciasis

*Onchocerca volvulus* a nematode living in in nodules under the skin of the patient is the causative agent for onchocerciasis. The nodules are large granulomas formed by tissue reaction around adult worms. Each nodule might host several worms of both sexes. The male worms have a length of 2 to 5 cm and the females 35 to 70 cm. The pathogenic organisms are microfilariae produced in numbers of 500 to 1500 per day from each female. The female, however, only can produce microfilariae if it lives symbiotic with bacteria belonging to the genus *Wolbachia*. The females are productive in about 10 years and the microfilariae live about 1 to 2 years. The microfilariae have a length of 300 µm. Severely infected patients may carry 2000 microfilariae in 1 mg upper dermis, whereas the microfilariae are seldom found in the blood or in other body fluids. Patients infected with *Onchocerca* nematodes suffer from dermatitis conditions such as leopard skin and depigmentation. These alterations of the skin are caused by reactions to dead or dying microfilariae [88]. Fibroblast proliferation leads to fibrosis and elastic fibers are replaced with hyalinized scar tissue. Some scars might also be caused by scratching. In advanced stages the skin resembles skin of very old individuals. Microfilariae might be visible in the cornea of patients. Dead microfilariae provoke inflammation leading to punctuate (snowflake) keratitis. Chronic inflammation causes opacification of the cornea. If not treated onchocerciasis might provoke optic nerve atrophy and chorioretinitis leading to loss of sight, hence the name river blindness [88,89]. The disease was first described as craw-craw in 1875 [88,89].

The life cycle (Figure 14) of the parasite involves as a vector blackfly belonging to the genus *Simulium*. The fly becomes infected with microfilariae when taking a blood meal. A new blood meal from another person might transfer the parasites to this person. The female flies taking blood meals restrict their flight within a few km from rivers. Thus, the transmission is limited to areas near rivers explaining the name river blindness. The original hotspots of the disease were river banks in Africa and Yemen. The slave trade probably also brought the disease to Central and South America [88]. A particular nasty consequence is that the disease makes the fertile soils often found on river banks inhabitable. The importance of *Simulium* blackflies as vectors was described in 1927 [89].

More than 99% of persons infected with onchocerciasis are found in Sub-saharan Africa. Isolated foci are found in Latina America and in Yemen. The transmission has been stopped in Ecuador, Columbia, Mexico and parts of Guatemala [88].

### 3.1. Ivermectin

For decades scientists at the Kitasato Institute, Japan, have grown unusual microorganisms and isolated new compounds from the culture media leading to the discovery of the antimicrobial pyrindicin, the protein kinase C inhibitor staurosporin and the proteasome inhibitor lactacystin. In 1973s. Omura became head of the Kitasato Institute and initiated a collaboration with MSD Research Laboratory, At the Kitasato Institute extraordinary microorganisms were cultivated and bioactivity of isolated compounds screened for in vivo bioactivity. The most promising compounds were forwarded to MSD research laboratory for in vivo tests. A product isolated from the broth of the soil bacteria *Streptomyces avermectinus* was tested in mice infected with the nematode *Nematospiroides dubius*. The product showed excellent anthelmintic activity with no toxic effects on mice. The active principles turned out to be eight avermectins (Figure 15, **42**–**49**). The most active compounds appeared to be the compounds of the B-series containing a 5-hydroxyl group [90].

Reduction of the C-22-C-23 double bond improved the activity. A mixture of 80% hydrogenated avermectin B_1a_ and B_1b_ was named ivermectin (**50**). The mixture possesses a broad activity against a broad range of nematodes. The agent killed external and internal parasites in horses, cattle, pigs, sheep and heartworms in dogs. The uncompelled anti-parasitic effects led to creation of a new term endectocide to describe compounds, which are capable of killing a wide variety of parasitic and health-threatening organisms both inside and outside the body [90,91,92]. Ivermectin interacts by preventing closure of glutamate and to a minor extent GABA-gated ion channels leading to hyperpolarization of the neural membrane and eventually killing of the parasite. The adult nematodes in general are not killed by the drugs. However, after administration of the drug female worms are prevented from releasing pathogenic amounts of microfilariae for 12 months even though the half live of the drug in humans is only 12 to 36 hours. GABA and glutamate receptors only occur in the central nervous system in mammalian but are commonplace in insects, nematodes and ticks. Ivermectin is unable to penetrate the blood-brain barrier in mammals, explaining the selectivity of the drug [90]. Ivermectin causes microfilariae to disappear rapidly from the skin in patients suffering from onchocerciasis. Apparently ivermectin prevents disarming the immune system enabling the host to kill microfilariae. The drug has to be taken for 18 years since even though the production of microfilariae is suppressed the adult worms are not killed. Ivermectin removes the symptoms of onchocerciasis since microfilariae are removed from the skin and after some time from the eye preventing further damage [90]. 

Additional anthelmintic macrolactones, the milbemycins, were found in the growth medium of *S.hygroscopicus*. The milbemycins possess the same macrocyclic lactone skeleton as avermectins but are not oxygenated at C-13 and consequently they are not glycosides [93]. 

#### 3.1.1. Macrocyclic Latones Veterinary Use

Since 1981 Merck & Co. has marketed a number of macrolactone formulations for veterinary use. Some analogues like eprinomectin (Figure 16, **51**) [94] and selamectin (**52**) [95] (Revolt, Zoaetis) are used for oral as well as topical application. Selamectin is a semisynthetic derivative prepared by deglycosylation of doramectin, hydrogenation of the C-22-C23 double bond, transforming C5 to a ketone and reacting it with hydroxylamine [96]. In contrast to ivermectin selamectin has no toxic effects on dogs and cats. Eprinomectin is not found in the milk from cows. Macrocyclic lactones of the ivermectin type have become the most used drugs for treatment of parasites in cattle, sheep, and pets in USA and UK [91]. In depth description of the different macrocyclic lactones, use and properties have been presented by Vercruy and Rew [97].

#### 3.1.2. Ivermectin for Human Use

After approval of the drug by the French Medical Agency in 1987 the drug was donated free of charge under the name of Mectizan by Merck & Co. Inc. (Fort Kenneworth, NJ, USA) to NGO organizations for treatment of onchocerciasis in as long as needed in amounts as much as needed [90]. The drawback of ivermectin is that treatment for keeping patients free from symptoms has to last for up to 18 years. Estimates of the number of infected persons in 1995 varied from 17 to 35 million cases globally. Not less than 0.27 million patients were blinded with 99% in in Sub-Saharan Africa [98,99]. Massive clinical trials in Africa have revealed that only sporadic side effects are observed and that the drug only needs to be administered orally once a year to keep the patient free from symptoms and to interrupt transmission. 

The African Program for Onchocerciasis Control was established in 1995 to administer mass drug administration to 19 African countries. The program was closed in 2015. The number of patients was in 2017 estimated to 1.34 million [98]. The final evaluation stated that onchocerciasis had been bee reduced in all African countries including conflict areas. Success of the program has been obtained in Sudan, Mali, Senegal, Burundi Malawi, Nigeria and Ethiopia where onchocerciasis is almost eliminated [99]. In western Uganda onchocerciasis was observed among 88% of persons older than 19 years in 1991 in Kabarole and Kyenjojo districts. An investigation in 2012 found no positive cases [99,100]. The activities have been taken over by WHO African Region Expanded Special Program for Elimination of Neglected Tropical Diseases. The program is estimated to prevent 17.4 million from suffering from the symptoms of onchocerciasis. Combined use of mectizan and vector control afforded better results than only distribution of drugs [99]. The Onchocerciasis Elimination Program for the Americas has succeeded in eliminating the disease in Amerika except in a cross-border region deep in the Amazon forest between Venezuela and Brasilia. About 30,000 people are at risk. The mass drug administration of ivermectin thus has severely reduced the burden of onchocerciasis [89,99]. Appearance of resistance have called for new combination therapies including ivermectin [101].

#### 3.1.3. Moxidectin

In 2018 FDA approved moxidectin (Figure 17, **53**) for treatment of onchocerciasis [69]. 

Moxidectin is a semisynthetic milbemycin. Milbemycins are not oxygenated at C-13 and consequently the compound does not possess the disaccharide side chain of avermectins. The absence of the carbohydrate moiety makes the compound more lipophilic. [93,98,103]. Moxidectin prepared from nemadectin isolated from the broth of *Streptomyce cyaneogriseus* subsp. *Noncyanogenous.* The 5-hydroxyl group of nemadectin is protected, the 23-hydroxy group oxidized, the methoxime introduced and the 5-hydroxy protecting group removed [103]. Moxidectin has been approved for treatment of onchocerciasis in humans and possesses a stronger and longer lasting suppression of microfilarial expression than ivermectin. Consequently it might be a better tool for elimination of onchocerciasis [98].

## 4. Lymphatic Filariasis

The pathogenic organisms in lymphatic filariasis caused are the nematodes *Wuchereria bancrofti, Brugia malayi* or *B. timori*. Lymphatic filariasis is characterized by lymphedema. After some years the edema may become non-pitting with thickening of the skin and loss of skin elasticity. Further progression leads to elephantitis [88] characterized by large disabling edemas. Over a billion people in more than 80 countries are threatened by this disease. The causative nematode *W. bancrofti* is found in tropical regions lf Africa, Asia, the Americas and the Pacific. The disease is particular frequent in hot and humid climates. *B. malayi* dominates in South-east Asia and South-west India. *B. tumori* is limited to islands in eastern Indonesia [88]. The life cycle of the parasite is similar to that of *Onchocerca* nematodes. The worms are found in lymphatic tissue of the human host. The female *W. bancrofti measures* 100 mm long and 0.25 mm in diameter. The microfilariae are released into the lymphatic vessels and from here they enter the veins. Female mosquitoes ingest the microfilariae and transmit them to a new human host by taking a new blood meal. The principal vector for *W. bacrofti* is *Culex quinquefasciatus.* In Africa *Anofeles* sp. also can transmit *W. bancrofti*. *B. malayi* uses bot *Mansonia* and *Anopheles* sp. as vectors. The time of development into the mosquito is about 12 days. In the human host the nematode might live and produce microfilariae for 20 years but the average lifespan is much shorter. The microfilariae have a lifespan of about 1 year. In the blood the density of microfilariae might reach 10,000/ml [88]. 

In addition to having reduced the burden of onchocerciasis ivermectin also is efficient in treating the parasitic diseases lymphatic filariasis. Ivermectin was approved for treatment of lymphatic filarisasis in 1998 by the French Medical Agency. Combination of albenzol and ivermectin was found 99% effective in elimination microfilaria from the blood. After GSK donated albenzol to the society and Merck extended the ivermectin donation to include lymphatic filariasis WHO initiated the Global Program to Eliminate Lymphatic Filariasis. From 2000 to 2020 7.7 billions doses against lymphatic filariasis have been distributed to 910 million persons in 68 countries reducing the number of people needing treatment from 1420 million to 597 million [104]. In the Americas Costa, Suriname, Trinidad and Tobago were removed from the WHO list of countries endemic for lymphatic filariasis in 2011 and it is hoped that Brazil, The Dominican Republic, Haiti and Guyana soon can be removed. WHO today recommends treatment with combination therapy consisting of ivermectin 200 µg/kg, diethylcarbamzine citrate (6 mg/kg) and albendazole 400 mg. In 2019 WHO has launched the ambition that onchocerciasis is eliminated in 2030 [98]. Sathoshi Omura and William C. Campbell together with You-you Tu shared the 2015 Nobel Prize in Medicine, the first two being mentioned for their work on ivermectin.

## 5. Cancer Diseases

A number of hallmarks are characteristic for cancer cells: (1) sustained proliferative signaling, (2) evading growth suppressors, (3) replicative immortality, (4) activated invasion and metastasis, (5) induction of angiogenesis, (6) resistance against cell death, (7) deregulation of cellular energetics and (8) not sensitive to immune destruction [105]. High rate of proliferation is characteristic for a number of cancer diseases and some of these are controlled by drugs targeting cells in the proliferative state. Spindle toxins offered a new efficient way of treating some cancer diseases like lymphomas and leukemia [106].The first spindle toxins vincristine and vinblastine were only discovered about 1960 [11]. 

### 5.1. Vinca Alkaloids

The leaves of Madagascan periwinkle [*Chataranthus roseus* (L.) G.Don Apocynaceae, named *Vinca rosea* by Linnaeus, Figure 18)] were used to control diabetes mellitus in traditional medicine on Jamaica. 

A product named vinculin had been marketed in England for treatment of diabetes. Injection of extracts from the leaves showed no antidiabetic activity, but, unexpectedly, killed rabbits [107]. Autopsy revealed multiple abscesses, particular in the liver and kidneys [107]. The rabbits died from infections becoming fatal because of a suppressed immune system. The white blood count was severely reduced and neutrophil leukocytes had virtually disappeared. In addition the bone marrow was destroyed [108]. It was realized that the active principle was partitioned into an acidic aqueous solution from a benzene extract. Chromatographic separation on almost neutral alumina afforded a brown mixture, from which a crystalline hydrosulfate was obtained. Injection of a solution of the crystals into rats caused a severe drop in the number of granulocytes after 36 hours, which lasted for three to six days [107]. Structural elucidation revealed that he compounds belonged to a previously unknown type of dimeric indole alkaloids [109]. The relative configuration of vinblastine Figure 19, **54**) and vincristine (**55**) was published in 1964 [110] and confirmed by an X-ray analysis [111], which also revealed the absolute configuration. Among the 120 alkaloids present in the leaves only two are used in approved drugs [112]. Three semisynthetic analouges also have been approved vindesine (**56**), vinorelbine (**57**) and vinflunine (**59**) [11].

Inspection of the structures of all the clinical used alkaloids reveals that they all possess the same carbon skeleton. Vincristine (**55**) is a formamide of dihydroindole, and vinblastine (**54**) is the *N*-methyldihydroindole. A formation of a bond from the α-methylester carbon of catharanthine (Scheme 5, **60**) and the carbon *ortho* to the methoxy group of vindoline (**59**) created the dimeric carbon skeleton. The biosynthetic reaction is catalyzed by α-3’.4’-anhydrovinbalse synthase (Scheme 5) [22]. The dimerization is a complex reaction, which is described in details elsewhere.

#### 5.1.1. Clinical Use

The ability to reduce the white blood count inspired researchers to investigate the chemotherapeutic potential of the alkaloids. Preclinical and clinical trials afforded overwhelming results and soon vincristine was introduced as combination therapy for treatment of childhood acute lymphoblastic leukemia (ALL) and non-Hodgkin’s lymphoma (NHL). The alkaloid is particularly effective in the treatment of hematological and lymphatic neoplasms, as well as of solid tumors (i.e., breast cancer, non-small cell lung cancer) [11]. Vincristine (**55**) is used for treatment of Philadelphia chromosome-negative acute lymphoblastic leukemia, β-cell lymphomas, metastatic melanoma, estrogen-receptor negative breast cancer, glioma, colorectal cancer, non-Hodgkin’s lymphoma, Hodgkin’s lymphoma, neuroblastoma, rhabdomyosarcsoma, multiple myeloma and Wilm’s tumor [11]. A combination of vincristine (**55**) and actinomycin C for treatment of Wilm’s tumor increased 2-year relapse free survival from 55% to 81% of patients with advanced disease [113]. Vincristine (**55**) is not as efficient for treatment of cancer diseases in adults as among children. However, it has some effect towards non-Hodgkin’s lymphoma and Hodgkin’s disease. In general the effect towards hematological malignancies are better than those against in solid tumors [113]. In particular neurotoxicity forms an upper limit for doses of vincristine (**55**) [113]. The convincing effects of vincristine for treatment of hematological malignancies particular in the childhood have led to characterization of the drug as a wonder drug [114].

The main side effects of vincristine (**55**) are neurotoxicity causing polyneuropathy [113]. Many signs of vincristine intoxication disappear weeks or months after termination of treatment. However, long term effects and even permanent damages are described [113]. The presently obtained experiences with the semisynthetic vinflunine (**58**) suggest that the side effects of this drugs are easier to manage [11]. 

Vinblastine (**54**) is used for treatment of leukemia, non-Hodgkin’s and Hodgkin’s lymphoma, breast cancer, nephroblastome, Ewing carcoma, small-cell lung cancer, testicular carcinoma and germ cell tumors [11]. 

#### 5.1.2. Mechanism of Action

Microtubules are key components of the cytoskeleton. They are crucial in the development and maintenance of cell shape, in the transport of vesicles, mitochondria, cell signaling and in mitosis [114]. The microtubules are polymers of α- and β-tubulin arranged in filamentous tubes. Both tubulins have a molecular weight of 55 kDa. The tubules might be many µm long [114]. Microtubule dynamics means that tubules are assembled and disassembled constantly [113]. During mitosis, first the chromosomes attach via their kinetochores to the spindle during prometaphase. Second complex movements of the chromosomes bring them in properly aligned positions for separations and finally during the anaphase microtubules drag each of the duplicated chromosomes towards the spindle poles. In the telophase they are incorporated in the two daughter cells (Figure 20) [114]. The spindle toxins or microtubule targeting agents inhibit microtubules function and thereby prevent the cell going from metaphase into anaphase. A prolonged senescence G1-state causes death of rapidly dividing cells by apoptosis [11,113,114]. The different spindle toxins bind to different binding sites in the tubulins. Vinblastine binds to the β-tubulin at the vinca-binding domain [11]. In low but clinical relevant concentrations the mitosis is prevented but at higher concentrations de-polymerization is observed [114]. 

#### 5.1.3. Analogues of Vincalkaloids

At the present five vinca alkaloids are in clinical use: the two natural products vinblastine (**54**) and vincristine (**55**) [115], and the semisynthetic compounds vindesine (**56**), vinorelbine (**57**), and vinflunine (**56**) [11]. Vindesine (**56**) is only approved in some countries, where it is used for pediatric solid tumors, malignant melanoma, blast crisis of chronic myeloid leukemia, acute lymphocytic leukemia, metastatic colorectal-, breast-, renal- and esophageal-carcinomas. Vinorelbine (**57**) has a wide antitumor spectrum of activity such as advanced breast cancer, advanced/metastatic non-small-cell lung cancer and rhabdomyosarcoma. The newest approved analogue vinflunine (**58**) is used for metastatic and advanced urothelial cancer after failure of platin-containing therapy [11]. In order to avoid resistance the vinca alkaloids are preferentially used in combination therapy with drugs possessing a different mechanism of action. Such drugs could be cychlophosphamide, an alkylating agent, doxorubicin and bleomycin both interacting with DNA, and cisplatin causing DNA damage [11]. In order to improve the effect new formulations such as incorporation into microspheres, nanoparticles and liposomes are being developed. In addition, new prodrugs are developed to target the agents toward the tumors [4,11,116].

#### 5.1.4. Sustainable Production of Vinca Alkaloids

It takes 2 tons of leaves to produce 1 g of vincristine and 500 kg to produce 1 kg of vinblastine (**54**). It is estimated that the production of 1 kg of vinblastine costs 1 million dollars and 1 kg of vincristine costs 3.5 million [112]. The extreme costs of production have provoked several attempts to produce the alkaloids by cell cultures. However, extensive metabolic cross talks and shuttling of about 35 intermediates synthesized via 30 enzymatic reactions in four different types of tissue and five different sub-cellular compartments have prevented the development of commercially interesting biotechnological production methods [117]. Hopefully a breakthrough can afford a feasible method for kg scale production of vinca alkaloids [115,118].

## 6. Conclusions

Natural products and natural products chemistry have had an impressive impact on drug development and consequently on our society [3]. An outstanding example of this is the fight against malaria. Until the discovery of quinine mankind had no efficient drug to control malaria, which was a devastating disease putting a severe burden not only on the population in tropical Africa but also on Europe and Asia [12,29]. It might be considered as a irony of history that the breakthrough to solve the problem of malaria was found in South America, where malaria was not a problem [33,35]. Open minded studies revealed that the bark of trees used to prevent muscle shivering turned out to be a warhead to combat malaria. Botanical studies let to the discovery of the mother trees, and a new genus, *Cinchona*, was created to enabling scientific description of the species with the healing bark [34,35]. A recipe for the use of the bark as a drug against malaria, Schedula Romana, was developed [34]. However, besides the unpleasant swallowing of the crude bark, an additional severe drawback of the bark was the variation of the level of the active principles. Isolation of the active alkaloid, quinine, in a pure state enabled for the first time development of an efficient drug with reproducible effects [31,49]. As with many other natural products development of a sustainable production of quinine became an absolute requirement for successful use of the drug. Until the Japanese occupation during the Second World War the sustainable production was maintained by plantations on Java and to a smaller extent in India. When the supply from Java was discontinued scientists developed synthetic antimalarial drugs, in particular chloroquine, which actually outcompeted quinine [57]. The development of chloroquine, however, was not the final stroke against malaria. After a few decades the parasites developed resistance and new drugs were desperately needed. In spite of all the disasters caused by the Vietnam War military need for a new drug led to the discovery of the artemisinins [80]. Artesunate and artemether are used in combination with other antimalarial drugs to postpone development of resistance. However, reports of lower sensitivities of parasites for artemisinin combination therapy are reported [85].

Onchocerciasis made fertile riverbanks in Africa inhospitable because of the threat of river blindness. The search for a drug to combat parasites in livestock led to the discovery of the macrocyclic lactone ivermectin [92]. Ivermectin is a very efficient drug for elimination of parasites in animals. The pathogenic agents in onchocerciasis in animals as well as in humans are *Onchocerca* nematodes. Fortunately ivermectin was found to be efficient for removal of the symptoms not only in animals but also in humans. Under the name of mectizan ivermectin is now offered free of charge for mass distribution in Africa and other parts of the world to combat onchocerciasis [90,99]. Even though impressive progresses have been made for relieving the world for the burden of onchocerciasis is still not eliminated but hopefully new drugs such as moxidectin and continued mass distribution will further reduce the burden [98].

Beside parasitic diseases cancer causes mortality and reduced life quality for a large part of mankind [119,120]. Until the 1960s hematological malignancies particular in the childhood had a high death rate [114]. Attempts to understand the antidiabetic effects of leaves from *C. rosesus* revealed that active agent had a poor effect on diabetes but severely suppressed the number of white blood cells. Intensive studies revealed that the active principles were spindle toxins and this opened a new path for treatment of diseases caused by fast proliferating cells [11,108]. Further studies revealed the structures of the compounds and enabled large scale isolation. However, as with many other natural products, sustainable supply in an economic feasible way is still an important restriction for the use of the compounds. Maybe studies may enable heterologous methods for production of the compounds in cell cultures [112,115,117]. Today many cancer diseases for which no cure existed before discovery of the spindle toxins are cured efficiently. 

The importance of natural products chemistry has been revealed by discovery of drugs with new mechanisms of actions like the spindle toxins for treatment of cancer diseases, the trioxolanes for treatment of malaria and the macrocyclic lactones for treatment of parasitic diseases. Many new drugs developed by inspiration from traditional uses turns out not have a curative effect on diseases for which their mother plants are used [33,35,108]. Open minded researchers realized that the active principles enabled treatment of incurable diseases, as in the case of quinine and vinca alkaloids. A drawback of natural products, however, is their limited supply. Sustainable production on a ton scale may enforce the use of alternative sources. Bioengineering of other organisms offers alternative possibilities [121] but are only possible if the biosynthesis is known in details [86,122]. 

In some cases preparation of derivatives of the natural product affords optimized drugs. This has been the case for the artemisinins like artemether (**33**) and artesunate (**35**) [7,81], macrocyclic lactones like moxidectin (**53**) [93] and vinca alkaloids like vindesine (**56**), vinorelbine (**57**), and vinflunine (**56**) [11]

Modern techniques have now enabled cultivation of fungi, marine organisms, endophytes and bacteria, which might open new paths for discovery of new families of natural products [5,6,123,124]. 

Above all an absolute requirement for new wonder drugs is devoted interdisciplinary research. As illustrated in the cases of vinca alkaloids and artemisinin, the isolation of an active principle is only the first step on the long and complicated path to registration of a new drug.

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
