# Peer review of "Natural Products That Changed Society"

_biomedicines, 2021, doi:10.3390/biomedicines9050472_

Round 1

Reviewer 1 Report

Dear Author

The submitted manuscript (biomedicines-1164597) describes the history of several natural compounds used to treat diseases with great health impact such as malaria, onchocerciasis and cancer.

The subject is interesting and feet the scope of biomedicines journal

I propose some changes/suggestions showing points where the manuscript can be improved.

1- The abstract should describe in more detail what will be presented and discussed in this review and not just an introduction to the topic.

2- Point 1 of the manuscript (introduction) seems out of context. How does this text, mostly on various theories on the synthesis of organic compounds, contribute to the introduction of the theme of natural products (not synthetic compounds) that changed the world? An introduction on one of the first highly successful medicines inspired by nature (aspirin is well included) and also on the diseases (origin, mechanism, worldwide distribution ...) combated by the medicines that the manuscript describes (quinine, artemisinine ...) would be more relevant.

3- Synthetic compounds that are not derived from natural structures should have less emphasis in this manuscript.

4- recently published review articles involving some of the natural compounds shown in the manuscript should be cited (artemisinine on https://doi.org/10.3390/foods10010065; vincristine on https://doi.org/10.3390/ijms19010263)

5- The point four (Lymphatic filariasis) is very little explored. Compared to points 2 and 3, little is said about the disease in question, its distribution, symptoms and history of the discovery of the effect of the natural medicine against the nematode causing the disease. As ivermectin is the only natural product indicated, the authors may be able to give an insight into the clinical studies taking place involving the drug in the treatment of Lymphatic filariasis disease.

6- the authors should correct the following typos

Line 16: vincristine instead vincirstine

Line 24: …limitations of this this paradigm came when… instead “…limitations of this this paradigm came when…”

Line 39: “…aspirin before 1899…” instead “…aspirin before 1900…”

Line 646: “…the white blood…” instead “…the whit blood…”

Author Response

Thank for a positive criticism of the manuscript. The reviewers comments have led to the below mentioned changes in the manuscript:

  1. The abstract has been rewritten and so it is in more detail emphasized what will be presented in the manuscript.
  2. The introduction has been rewritten starting now with the natural products discussed in the manuscript. In addition the burden of some of the diseases has been included in the introduction. The Introduction, however, still discuss organic synthesis since the interaction between organic synthesis and natural products chemistry is fundamental for understanding the state of art of drug development.
  3. Synthetic compounds have been included to the extent that is needed for understanding the interaction of natural products chemistry and natural products chemistry.
  4. The two reviews have been included in the reference list.
  5. The paragraph on lymphatic filariasis has been elaborated.
  6. The typos have been corrected.

Reviewer 2 Report

 I am pleased to recommend publication of this manuscript in Biomedicines. It is very well written and informative, collecting information scattered in terms of time and sources. Complimenting to the author, I only have minor issues to highlight for a revision:

  1. Page 6, line 202. I suggest to replaced “washed” with “extracted”, since quinine goes into the acidic phase, that is not discarded.
  2. Page 9, lines 307-311. Regarding the apicoplast genome, it would be also interesting to report that, although an animal, Plasmodium has an active MEP pathway for the synthesis of isoprenoids, and is, indeed, sensitive to its inhibitors.
  3. Page 13: line 380. I believe that it is hemozoin (compound 30) and not heme (28) that precipitates
  4. General comment on the synthesis of artemisinin. I suggest to mention that artemisinin was first isoilated in former Yugoslavia (see Nat.Prod.Commun. 2017, 12, 1157)5. Page 16. I would number the carbons referred to in the text in the formula of avermectins.6. Page 18: line 576. The formation of the oxymino group involves previous oxidation of the the 24-hydroxyl.7. Page 20: Scheme
  5. The dimerization of catharantine and vindoline is seemingly oversimplified. It is a complex reaction, triggered by a Potier-Polonowski rearrangement.8. Page 23, line 736. Is the Latin term “Schecula Romana” correct?Overall, a very nice article

Author Response

Thank for a positive criticism of the manuscript. The reviewers comments have led to the below mentioned changes in the manuscript:

  1. Washed has been replaced with extracted.
  2. A sentence has been included in 2.1.6 that attempts to target the apicolast have been successful with tetracycline but no with fosmidomycin.
  3. It now appears from the text that hemozoin precipitates..
  4. I appreciate that the reviewer has called my attention to the illustrative story of early isolation of artemisinin. It has now been included in 2.1.8.
  5. Numbering has been included in fig. 15
  6. The semisyntheis of selamectin is now described.
  7. A reference to the mechanism of dimerization of vinca alkaloids have bee added.
  8. Schecula has been corrected to schedula